# Cohort profile: an observational longitudinal data collection of health aspects in a cohort of female cancer survivors with a history of pelvic radiotherapy—a population-based cohort in the western region of Sweden

Linda Åkeflo [1], Gail Dunberger,[2] Eva Elmerstig,[3] Viktor Skokic,[1] Gunnar Steineck,[1] Karin Bergmark[1]

**To cite:** Åkeflo L, Dunberger G, Elmerstig E, *et al*. Cohort profile: an observational longitudinal data collection of health aspects in a cohort of female cancer survivors with a history of pelvic radiotherapy—a population-based cohort in the western region of Sweden. *BMJ Open* 2021;**11**:e049479. doi:10.1136/bmjopen-2021-049479

For numbered affiliations see end of article.

**Correspondence to**
Mrs Linda Åkeflo;
linda.akeflo@vgregion.se

## ABSTRACT

**Purpose** The study 'Health among women after pelvic radiotherapy' was conducted in response to the need for more advanced and longitudinal data concerning long-term radiotherapy-induced late effects and chronic states among female cancer survivors. The objective of this paper is to detail the cohort profile and the study procedure in order to provide a sound basis for future analyses of the study cohort.

**Participants** Since 2011, and still currently ongoing, participants have been recruited from a population-based study cohort including all female patients with cancer, over 18 years of age, treated with pelvic radiotherapy with curative intent at Sahlgrenska University Hospital in Gothenburg, in the western region of Sweden, which covers 1.7 million of the Swedish population. The dataset presented here consists of baseline data provided by 605 female cancer survivors and 3-month follow-up data from 260 individuals with gynaecological, rectal or anal cancer, collected over a 6-year period.

**Findings to date** Data have been collected from 2011 onwards. To date, three studies have been published using the dataset reporting long-term radiation-induced intestinal syndromes and late adverse effects affecting sexuality, the urinary tract, the lymphatic system and physical activity. These projects include the evaluation of interventions developed by and provided in a nurse-led clinic.

**Future plans** This large prospective cohort offers the possibility to study health outcomes in female pelvic cancer survivors undergoing a rehabilitation intervention in a nurse-led clinic, and to study associations between demographics, clinical aspects and long-term late effects. Analysis focusing on the effect of the interventions on sexual health aspects, preinterventions and postinterventions, is currently ongoing. The cohort will be expanded to comprise the entire data collection from 2011 to 2020, including baseline data and data from 3-month and 1-year follow-ups after interventions. The data will be used to study conditions and treatment-induced late effects preintervention and postintervention.

### Strengths and limitations of this study

► The major strength of the study is the large representative population-based cohort (n=605).
► Uses a large longitudinal dataset consisting of female cancer survivors' entries from 6 months to several years post pelvic radiotherapy treatment.
► Highlighting treatment-induced cancer survivorship diseases and chronic states improves the possibilities for developing further effective treatments.
► One limitation of the study is the reliance on self-reported data, which has the potential for response bias.
► The interventions that were provided varied over the course of the study, which is considered a potential limitation.

## INTRODUCTION

An increasing number of individuals now live longer after a cancer diagnosis than was previously common, and more and more patients are living with treatment-induced late effects, treatment-induced cancer survivorship diseases and survivorship chronic states.[1–3] Pelvic radiotherapy affects the intestinal and urinary tracts, the lymphatic system and sexual health.[4–10] In a video at https://www.jostrust.org.uk/video/letstalk-pelvic-radiation-disease-after-cervical-cancer, some of more than 20 million cancer survivors in Europe describe how radiation-induced impaired intestinal health affects them. In Sweden, approximately 3500 women are diagnosed with pelvic cancer (gynaecological, anal or rectal cancer) every year.[11] In 2011, we started a nurse-led clinic focusing on treatment-induced late effects among pelvic cancer survivors with the intent of developing

self-care strategies and treatments. We have previously reported on the benefits of the clinic regarding the improved quality of life and psychosocial well-being among female pelvic cancer survivors.[12]

We aim to describe a population-based cohort consisting of female cancer survivors treated with pelvic radiotherapy using a dataset from a longitudinal study that has been ongoing since 2011. A further aim is to describe the data collection procedure, the interventions provided in a nurse-led clinic and the characteristics of the study cohort. In addition, a few basal empirical results will be presented to illustrate the study cohort. The objective of this paper is to detail the cohort profile and the study procedure in order to provide a sound basis for future analyses in the study cohort.

## COHORT DESCRIPTION
### Setting
In 2009, the Swedish government proposed a new regulation: 'A National Cancer Strategy' (SOU2009:1).[13] In collaboration with the Swedish county councils and regions, six Regional Cancer Centres (RCC) were established. Based on the national strategy proposals, the RCC in Western Sweden financed a nurse-led clinic at Sahlgrenska University Hospital. The nurse-led clinic was founded in 2011 by a senior consultant physician who has a PhD and is specialised in gynaecological oncology, and an oncology nurse with a PhD in oncology. The team currently consists of three clinical oncology nurses who are specially trained and clinically specialised in understanding and addressing issues concerning pelvic cancer survivorship late effects. The cancer survivors receive education about radiotherapy-induced late effects, basic anatomy and physiology and together with the nurse, decide on supportive care actions regarding medication, nutrition, and coping with sexual, psychological and social challenges. In cases with more specific needs, intercurrent diseases, or a suspected recurrence of cancer, the patient meets the senior medical consultant who has the primary medical responsibility for the clinic. The senior medical consultant meets regularly with the team and discusses plans for patients' medication with the staff member who will issue the prescriptions. The healthcare programme developed in the clinic is based on programmes developed by others and on our previous studies in this area.[4 14–20]

### Patient and public involvement
Patients and/or the public are not involved in the design, conducting, reporting or dissemination plans in this research. Results from future studies from the cohort will be submitted for publication in peer-reviewed journals and presented at relevant conferences.

### Participants
Study participants are recruited from two different cohorts: (1) a population-based study cohort group including all female patients with cancer treated with curative intent from 2007 onwards who are continuously identified from medical records at Sahlgrenska University Hospital in Sweden and (2) a referred patient group including all female patients with cancer referred to the rehabilitation clinic. Inclusion criteria are female cancer survivors who completed pelvic radiotherapy at least 6 months prior to inclusion. Exclusion criteria are cancer recurrence, inability to comprehend or answer the questionnaire, and poor proficiency in the Swedish language. Data are collected from female cancer survivors with a history of pelvic radiotherapy in Western Sweden, an area that includes 20% of the Swedish population. The tradition of collecting registry data in Sweden offers good opportunities for studying cancer survivors without selection-induced problems. The dataset contains information about the preintervention and postintervention needs of women who received pelvic radiotherapy at Sahlgrenska University Hospital between 2007 and 2016. Female cancer survivors referred to the nurse-led clinic are also invited to participate in the study.

### Invited women
An introductory letter is sent to eligible study participants. Shortly after, a research secretary phones them and gives oral information about the study, asks if they are willing to participate and, if so, asks them to give permission to be sent a written informed consent and the questionnaire. When the baseline questionnaire is returned, the participant receives an invitation to attend the nurse-led clinic where individual healthcare interventions are conducted. The code number from the questionnaire (followed throughout the data collection) is entered into the patient-database FileMaker Pro17 Advanced that was specifically designed to suit the study. The study participant receives a follow-up questionnaire three and twelve months after the completed intervention. The research secretary keeps track of each patient and, in case questionnaires are delayed or missing, sends two reminders at time points determined in advance (see figure 1).

### Referred patients
Patients referred from oncology healthcare providers, general practitioners and through private referrals, who meet the inclusion criteria are also invited to participate in the study. The data collection procedure follows the same principles as for the invited women in the population-based cohort.

### Study-specific questionnaires
As shown in table 1, the baseline questionnaire consists of 175 questions, divided into eight main sections and one concluding chapter. Information is obtained about the frequency, intensity, duration, and quality of each symptom, and the degree of distress it causes the patient. An example of one question is: 'how many times per week (approximately) have you had bowel movements during the past 6 months?' with possible answers: 'about

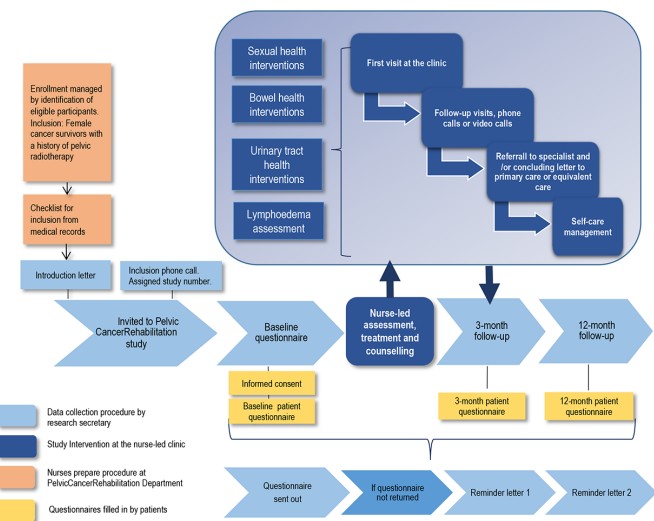

**Figure 1** Flowchart of pelvic cancer rehabilitation ongoing data collection procedure and the procedure for patient inclusion, contact with participants and the follow-up points for questionnaires.

every other day', 'less than once a day', 'once a day', 'up to two times a day', '2–3 times per day', '3–4 times per day' or 'five times or more per day'. To facilitate statistical analyses, the answers are coded into values, ranked and placed into groups. Variables such as age and number of children are categorised into groups.

### Several sections allow free comments
The questionnaire also addresses the extent to which the symptoms affect social functioning. In the concluding chapter, the participants are invited to visit the nurse-led clinic. Some questions require the patients to rank their most distressing symptoms. In the follow-up questionnaires, the study participants' health status and symptoms are measured, and the interventions conducted at the nurse-led clinic are evaluated.

### Diagnostic and alleviative means
The questionnaire serves as a basis for the interventions conducted at the first visit to the clinic. The patient's current health status reveals the areas that need to be focused on. Current symptoms and self-care strategies are discussed. New strategies are then suggested and decided on together with the patient. The duration of the contact and the number of follow-ups varies from 1 month to 1 year depending on the symptoms and the effect of treatments and interventions.

### Intestinal health
The assessment is based on five syndromes: urgency to defecate, faecal leakage, excessive gas production, excessive mucus discharge and blood discharge.[21] Symptoms, such as leakage of mucus and blood, abdominal bloating, signs of bacterial overgrowth and bile salt malabsorption, are also assessed. Diagnostic means, including blood tests such as electrolytes, blood counts, vitamin B12 and serum magnesium, are taken when indicated. An algorithm

developed by the research group serves as a guide in the clinical setting.[12] A mobile application has also been developed that was specially designed for radiation-induced intestinal syndromes and which can show graphs of the variation in intestinal health over time. The application serves as a complement to existing methods of patients' self-assessment and self-management, and helps to support the nurse's and patient's conversation and decision-making.[22 23] The objective of the interventions is to restore or improve intestinal health and includes medical treatment, pelvic floor muscle training and techniques using cognitive training to control intestinal function. Details of these interventions can be accessed elsewhere.[12 24]

### Sexual health
The PLISSIT-model (Permission, Limited Information, Specific Suggestions, and Intensive Therapy) developed by Annon[25] is used to address sexual health concerns. The first three levels in the model are helpful for most patients. When specific suggestions are not sufficient, the patient is recommended to take part in intensive therapy, which is sometimes given in the clinical setting and other times with a sexologist or psychotherapist after a referral is sent. Female sexual dysfunction is a complex condition, including diverse aspects, definitions and classifications.[26] To assess and treat sexual dysfunction, previous sexual practices and experiences are discussed, with particular attention given to patients' integrity, special needs and preferences. Structured information about common physical late effects involving vaginal changes and sexual problems is provided as well as information concerning psychological issues influencing sexual health; all information given is in accordance with evidence-based knowledge and practice in the field.[1 4 17 27–29] Patients receive information about and instructions on the use of vaginal dilator therapy and topical oestrogen, and they are also given guidance and suggestions related to lack of sexual desire. When requested, specific sexual devices, films and literature are suggested to encourage and help the patient to regain sexual function, which for some patients is helpful in managing issues of body image perception, self-esteem and intimacy.

### Sexual abuse
Patients with a history of sexual abuse have an increased risk of developing sexual problems after cancer treatment, and a large proportion of women with cervical cancer and those experiencing dyspareunia postcancer treatment have been sexually abused.[30] In cases where this is reported, experiences of sexual abuse are carefully and sensitively discussed with the patient and, in some cases, this may lead to referral to a psychologist or therapist.

### Urinary tract health
Symptoms of urgency, nocturia, urinary retention, urinary tract infection and pelvic pain are addressed. The diagnostic methods used include evaluations of urinary

**Table 1** The study-specific baseline questionnaire, divided into eight sections and one concluding chapter

| Section | Question areas |
|---|---|
| Sociodemographic | Gender<br>Age<br>Marital status<br>Number of children<br>Level of education<br>Employment status |
| Quality of life and well-being | Quality of life<br>Depression<br>Worry<br>Anxiety |
| Body perception and self-image | Femininity<br>Self-esteem<br>Fertility<br>Childbirth<br>Vaginal and perianal injury related to childbirth or physical trauma |
| Intestinal and defecation habits | Loose stools<br>Faecal incontinence<br>Urgency to defecate<br>Excessive gas<br>Abdominal pain<br>Having a stoma<br>Medical treatment related to intestinal symptoms |
| Micturition habits and urinary tract symptoms | Urinary frequency<br>Urgency to urinate<br>Nocturia<br>Having a urinary catheter<br>Medical treatment due to symptoms |
| Sexual health | Menopause<br>Use of systemic hormone replacement therapy<br>Use of topical oestrogen<br>Impaired lubrication<br>Vaginal shortness<br>Vaginal inelasticity<br>Dyspareunia |
| Sexual abuse | Experience of sexual abuse<br>Experience of sexual harassment<br>Age when experienced sexual abuse or harassment<br>Extent to which the experience affects sexual life |
| Lymphoedema | Heaviness in legs, genitals and abdomen<br>If diagnosed with lymphoedema, current lymphoedema treatment |
| Concluding chapter | Self-reported needs<br>Invite to visit the nurse-led clinic for assessment and counselling |

frequency and of symptoms, which are divided into irritative symptoms, obstructive symptoms and bleeding. Patients can keep a 72-hour voiding diary, including urine volume and fluid intake, which is a helpful resource for this discussion. Guidance and recommendations are given about topical oestrogen, pelvic floor muscle training, medical treatments, self-care modifications and behavioural interventions, all in accordance with previous evidence-based knowledge.[31 32] In cases of severe urinary tract symptoms, referrals are sent to a urotherapist, physiotherapist, or urologist.

### Lymphoedema

Lower limb lymphoedema is a common non-curable chronic complication with multifactorial pathophysiology.[33] Early detection and treatment of lymphoedema are essential in order to prevent complications. Self-reported or objectively assessed swelling or heaviness in the lower limbs leads to referral for lymphatic therapy. The primary therapy is the use of individually tailored compression garments and second-line therapy consists of manual lymphatic drainage with intermittent use of pumps and other self-care strategies.[34 35]

### Statistical analyses

Initially, data from the questionnaires were entered into EpiDataSoftware V.3.1 (EpiData Association). Since only one answer to each question is allowed, in the event of two answers the procedure was to alternate entering the first provided answer and the second provided answer. In order to facilitate the analysis of responses to the open-ended questions, the answers to these questions were transcribed in Microsoft Word (2016). R V.3.5.2. was

used for statistical analysis of the data. The results will be reported in means, medians and percentages.

## FINDINGS TO DATE

Three studies based on the cohort have been published.[12 36 37] Two papers focused on gastrointestinal side effects and one on sexual health aspects. It was found that the majority of the women reported a change in bowel habits and, in almost half of them, the effect was considerable.[12] Women with faecal leakage were less likely to practise physical activity than survivors without leakage, and survivors who practised weekly physical activity experienced better quality of life and were less frequently in a depressed mood than women not physical active.[36] Furthermore, a statistically significant increase in psychological distress and sexual health impairment was found among women with a history of sexual abuse compared with women without such experience.[37]

### Characteristics of study participants

From January 2011 to June 2017, we identified a total of 791 patients in the population-based cohort who met the inclusion criteria. Of the total sample, 684 (86%) individuals gave oral consent to participate in the study and of these, 464 (68%) completed the baseline questionnaire.

During the same period, 184 referred patients met the inclusion criteria and were invited to participate in the study. Of those invited to participate, 141 (76.6%) completed the baseline questionnaire and of these, 131 (92%) took part in interventions.

The responses and completeness of the questionnaires from both study groups are shown in the schematic diagram (see figure 2). In table 2, we present the baseline characteristics and demographics in the dataset, which consists of 605 participants: 464 (76.7%) from the invited cohort and 161 (23.3%) from the referred cohort.

The majority of the study participants had a history of gynaecological cancer and had been treated with radiotherapy in combination with surgery. In the population-based study cohort, the mean age was higher and two times as many were retired compared with the referred group. In the referred group, two times as many were on sick leave. In total, 379 (63%) of the 605 study participants agreed to visit the clinic (see table 3). Sixty-seven (14.4%) of the women in the population-based study group who declined to visit the clinic reported that they had radiotherapy-induced late effects.

In this paper, we describe the population-based cohort consisting of female cancer survivors treated with pelvic radiotherapy in the western region of Sweden. The data

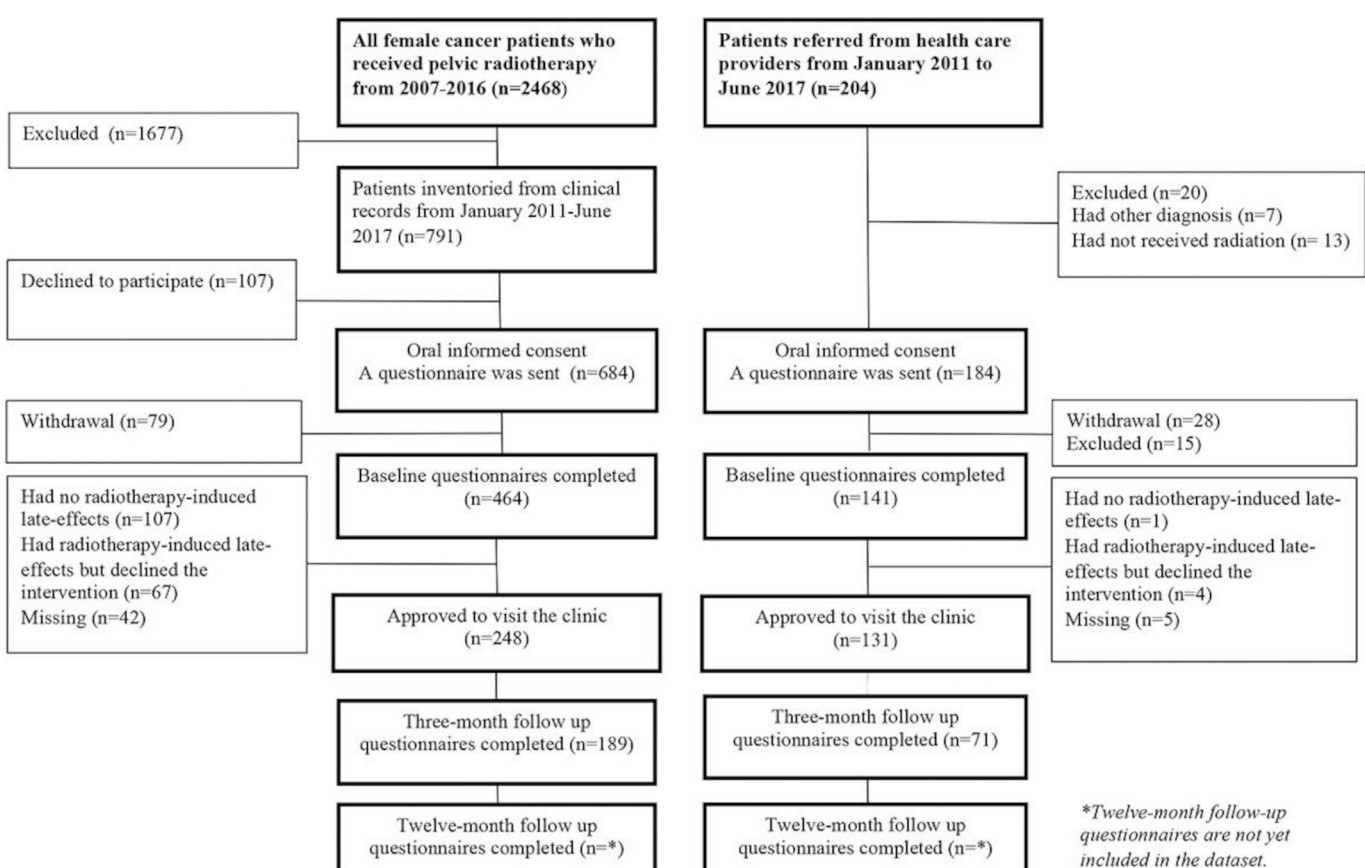

**Figure 2** Schematic diagram of data collected from two different study cohorts from January 2011 to June 2017: a population-based cohort and patients referred from oncology healthcare providers, general practitioners and through private referrals who met the inclusion criteria. The diagram includes study response rate, completeness of questionnaires and reasons for loss of participants.

**Table 2** Characteristics of the study participants

| Variable | Total | Invited | Referred |
|---|---|---|---|
| Participants, n (%) | 605 | 464 (76.7) | 141 (23.3) |
| Cancer type, n (%) | | | |
| Endometrial cancer | 216 (35.7) | 181 (39.0) | 35 (24.8) |
| Cervical cancer | 132 (21.8) | 80 (17.2) | 52 (36.9) |
| Ovarian cancer | 2 (0.3) | 1 (0.2) | 1 (0.7) |
| Vaginal cancer | 5 (0.8) | 3 (0.6) | 2 (1.4) |
| Vulvar cancer | 21 (3.5) | 19 (4.1) | 2 (1.4) |
| Anal cancer | 80 (13.2) | 58 (12.5) | 22 (15.6) |
| Rectal cancer | 145 (24.0) | 122 (26.3) | 23 (16.3) |
| Other | 4 (0.7) | | 4 (2.8) |
| Age in years | | | |
| Mean | 64.5 | 66.5 | 57.6 |
| SD | 12.6 | 11.5 | 13.6 |
| Missing, (%) | 11 | 7 (1.5) | 4 (2.8) |
| Years since radiotherapy, grouped | | | |
| 0 | 35 (5.8) | 6 (1.3) | 29 (20.6) |
| 1 | 219 (36.1) | 166 (35.8) | 53 (37.6) |
| 2 | 98 (16.1) | 86 (18.5) | 12 (8.5) |
| 3 | 139 (23.0) | 131 (28.2) | 8 (5.7) |
| ≥4 | 105 (17.3) | 69 (14.9) | 36 (25.5) |
| Missing | 9 (1.5) | 6 (1.3) | 3 (2.1) |
| Mean | 2.6 | 2.2 | 3.9 |
| SD | 3.4 | 1.2 | 6.7 |
| Cancer treatment, n (%) | | | |
| External radiotherapy, only | 145 (24.0) | 101 (21.8) | 44 (31.2) |
| External radiotherapy and brachytherapy | 20 (3.3) | 16 (3.4) | 4 (2.8) |
| External radiotherapy, brachytherapy and surgery | 180 (29.7) | 157 (33.8) | 23 (16.3) |
| External radiotherapy and surgery | 260 (43.0) | 190 (40.9) | 70 (49.6) |
| Marital status, n (%) | | | |
| Married or living with a partner | 402 (66.4) | 309 (66.6) | 93 (66.0) |
| Widow | 70 (11.6) | 59 (12.7) | 11 (7.8) |
| Has a partner but lives alone | 30 (5.0) | 15 (3.2) | 15 (10.6) |
| Single | 102 (17.0) | 80 (17.2) | 22 (15.6) |
| Missing | 1 (0.2) | 1 (0.2) | |
| Education level, n (%) | | | |
| Elementary school | 173 (29.1) | 150 (32.3) | 23 (16.3) |
| Secondary school | 227 (38.2) | 169 (36.4) | 58 (41.1) |
| College/university | 194 (32.0) | 135 (32.3) | 59 (41.8) |
| Missing | 11 (1.8) | 10 (2.2) | 1 (0.7) |
| Employment status, n (%) | | | |
| Student | 4 (0.7) | 3 (0.6) | 1 (0.7) |
| Unemployed job seeker | 12 (2.0) | 10 (2.2) | 2 (1.4) |
| Employed | 162 (27.0) | 116 (25.0) | 46 (32.6) |
| Housewife | 4 (0.7) | 2 (0.2) | 2 (1.4) |
| On sick leave | 54 (9.0) | 17 (3.7) | 37 (26.2) |

Åkeflo L, *et al. BMJ Open* 2021;**11**:e049479. doi:10.1136/bmjopen-2021-049479

| Table 2 Continued | | | |
|---|---|---|---|
| **Variable** | **Total** | **Invited** | **Referred** |
| Disability pension | 35 (5.7) | 26 (5.6) | 9 (6.4) |
| Retired | 328 (54.7) | 284 (61.2) | 44 (31.2) |
| Missing | 6 (1.0) | 6 (1.3) | |
| Resident, n (%) | | | |
| In a big city | 182 (30.6) | 131 (28.2) | 51 (36.2) |
| In a small or medium-sized city | 309 (50.2) | 244 (52.6) | 61 (43.3) |
| In the countryside | 116 (18.9) | 87 (18.8) | 29 (20.6) |
| Missing | 2 (0.3) | 2 (0.4) | |
| Smoking, n (%) | | | |
| Does not smoke | 448 (74.0) | 339 (73.1) | 109 (77.3) |
| Smokes | 67 (11.0) | 52 (11.2) | 17 (12.1) |
| Missing | 88 (14.5 | 73 (15.7) | 15 (10.6) |

N (number) and proportion (%) of women are presented.

collection procedure, the interventions provided and the characteristics of the study cohort are also outlined. In addition, a few basal empirical results are reported to illustrate the study population. The major strength of the study is the large population-based cohort since it creates the possibility of studying cancer survivors without selection-induced problems and makes it representative of the reference population consisting of an increasing number of female cancer survivors treated with pelvic radiotherapy. The longitudinal study design enables future investigations of long-term treatment-induced late effects, diseases and chronic states. Moreover, it will be possible to evaluate the long-term outcomes of interventions and the treatments provided.

The data collection has generated a large dataset consisting of patient entries from 6 months to several years post pelvic radiotherapy . Over a period of six and a half years, almost 1000 female pelvic cancer survivors have been invited to participate in the study; the dataset consists of 605 cancer survivors. In the population-based cohort group, 68% completed a baseline questionnaire. Treatment and interventions concerning physical, psychosocial and sexual issues were offered to patients in both the population-based cohort and the referred cohort, and these were evaluated. As shown in table 3, 53.4% of the study participants in the population-based cohort agreed to visit the clinic, which may indicate

the proportion of cancer survivors with unmet needs. Improved self-care strategies, increased clinical knowledge and developments in technology currently provide healthcare professionals with possible methods to help cancer survivors manage and treat the late effects of pelvic radiotherapy treatment. We believe that highlighting treatment-induced cancer survivorship diseases and chronic states may increase the likelihood of further effective treatments being developed.

Cancer survivorship issues have advanced from being neglected to gradually being given increasingly greater attention in healthcare practice as well as in research. This is reflected by the development of national guidelines and the organisation of national and international scientific conferences and meetings in this subject area. Efforts are being made to both understand how treatment-induced late effects manifest in pelvic cancer survivors and to find strategies to deal with these late effects. Twenty years have passed since researchers within our team[4] first observed that patients with gynaecological cancer suffered from vaginal changes affecting their sexual health. More recently, we have identified five syndromes impairing pelvic cancer survivors' intestinal health. Results published in 2017[21] simplify the search for ways to prevent, manage and help minimise the occurrence and intensity of survivorship diseases. One promising ongoing study being conducted by Schofield et al in Australia is

| Table 3 Number (N) and proportion (%) of study participants who agreed to visit the clinic | | | |
|---|---|---|---|
| | **Total N (%)** | **Invited N (%)** | **Referred N (%)** |
| No, I have no late effects and do not need to visit the clinic | 108 (17.8) | 107 (23.1) | 1 (0.7) |
| No, I have late effects, but I do not want to visit the clinic | 71 (11.7) | 67 (14.4) | 4 (2.8) |
| Yes, I want to visit the clinic | 379 (62.6) | 248 (53.4) | 131 (92.9) |
| Missing | 47 (7.8) | 42 (9.1) | 5 (3.5) |

N (number) and proportion (%) of women are presented.

evaluating a care programme similar to ours.[38 39] Furthermore, in 2015 Andreyev *et al*[15] published a guide for the management of intestinal problems, including an algorithm that, in our opinion, is a useful tool for clinicians. However, the pathophysiological changes, which are described in numerous previous studies, are not yet fully understood.[19–21] Inflammatory and fibrotic processes in the gut wall are probably of importance in explaining the various symptoms. The processes may relate to both the intestinal tract and other organs located in the pelvis. Hofsjo *et al* in our research group[40] recently studied vaginal changes and found morphological explanations for changes in the vaginal wall. Biopsies from the vaginal connective tissue affected by radiotherapy showed dense collagen and entangled elastin fibres, a finding that may explain common symptoms such as reduced vaginal elasticity during intercourse, reduced lubrication, and dyspareunia.

In Sweden, promising steps are being taken towards developing a programme for national coordination of cancer rehabilitation practice. Cerna *et al*,[23] in a study concerning self-management from an educational perspective, observed that patients and nurses can together create tailor-made solutions. The nurses focus on encouraging the patients to self-reflect and on maintaining the patient's motivation to continue to engage in self-care. The ongoing establishment of oncology nurse navigators[41] is also an initiative that has the potential to increase the supportive care given to cancer survivors. In the 1980s, research in the psychological field[42] showed that patients have a high risk of anxiety related to oncology treatment. Recent studies report that patients have a lower risk of future anxiety and depression when their needs are addressed during treatment.[43 44] In our opinion, when planning future follow-up in clinical cancer care, these findings need to be taken into account, irrespective of the patient's cancer diagnosis. We suggest that, in the future, healthcare should provide advanced specialist expertise in the management of severe treatment-induced late effects.

Sexual concerns are generally not addressed or discussed as much as patients would like,[45] and patients generally wait for healthcare professionals to raise the subject.[46] In our clinical setting, sexual health conversations are integrated into the clinical work, routines for assessment and treatment have been created, and it becomes clearer when a patient should be referred to specialists and sexologists. The PLISSIT-model[25] has been used for addressing sexual health concerns in both our own clinical setting and in that of others. Clinical experience shows that sexual function might improve and even return to prediagnosis level through frequent clinical follow-up, which is consistent with results from previous studies.[47] Parts of WHO's definition of sexual and reproductive health state that: 'sexual health requires a positive and respectful approach to sexuality and sexual relationships',[48] so healthcare professionals need to be able to talk openly to patients about sexuality. It has

been suggested that healthcare professionals should actively engage in training to improve their communication skills in order to overcome common communication barriers.[49 50] Our clinical experience has shown that several patients have felt relieved when being able to speak openly about sexual issues as well as other private health concerns. The baseline questionnaire seems to serve as a therapeutic tool that can be followed up in counselling.

One could argue that the observational study design is a limitation in this study and that a randomised clinical trial might provide data that are more reliable. However, for ethical reasons, we found it necessary to offer all patients in the population the best available intervention. Hence, we considered that the observational study design was best suited to this purpose, a design previously shown to produce reasonably useful results.[51]

One important limitation of the study is the reliance on self-reported data, which has the potential for response bias. However, we considered self-reported data to provide a wider range of responses than data collected using other data collection instruments.[52] To avoid information-related problems, we took advance preventive action by using questionnaires based on the clinometric method that we have introduced, developed, used and described in previous research projects.[4–6 53] We employed epidemiological methods, introduced into the cancer survivorship field by a hierarchical step-model, to manage bias and confounding.[54] This method was considered appropriate to measure causal relationship, and to examine different symptoms and characteristics in treatment-induced side effects. In short, the method comprises a quantitative prephase of semi-structured interviews with persons suitable for the study. Thereafter, a face-to-face-validation is conducted to ensure satisfactory internal consistency.

The wide range in length of time since completion of treatment may be considered a weakness in the study; however, this will probably also facilitate measurement of symptom progression in future analyses. Another potential limitation is that the interventions provided varied over the course of the study due to the increase in both ours and others' understanding of the complexity and pathophysiological mechanisms of symptoms and late effects. The minor changes in the interventions during the study need to be considered in future analysis. The results may also be affected by non-participation and loss to follow-up. We can only speculate about the reasons for these, such as patients not having time, not feeling motivated or being unwilling to recall their previous cancer experience. Possible reasons for patients declining an offer to visit the clinic despite having troublesome symptoms could be due to the symptoms being too severe to enable travel to the clinic. Age-related problems and long-distance transportation can be other reasons. The data collected from the referred patients will allow analysis of the prevalence of symptoms and unmet needs observed by other healthcare providers. Since the data

were collected in Sweden, we do not know to what extent our analysis will be applicable to other populations.

Through frequent lectures for patients and healthcare professionals, and the implementation of treatment strategies directly to patients, we apply the knowledge we have acquired and use it in the clinic and with the cancer survivors themselves. It is also worth mentioning that our rehabilitation clinic serves as a model for similar clinics in other regions of Sweden that have been established as part of the national strategy financed by the Swedish government. The current nurse-led clinic may serve as the beginning of a future tertiary centre to develop interventions and treatments for cancer survivors.

The extent to which the interventions provided in the individualised nurse-led rehabilitation might improve health in female cancer survivors treated with pelvic radiotherapy is currently unclear. The dataset from the 3-month follow-up questionnaire was prepared and used in recently published studies, while preparation of the dataset from the 1-year follow-up questionnaire is ongoing. To the best of our knowledge, our study cohort is one of only a few published population-based cohorts of female pelvic cancer survivors with treatment-induced late effects receiving individualised interventions with a focus on physical and sexual health after radiotherapy. The interventions developed, outlined and provided will hopefully contribute to further development of evidence-based management strategies in pelvic cancer rehabilitation and will be reported in future papers.

**Author affiliations**
[1]Division of Clinical Cancer Epidemiology, Department of Oncology, Institute of Clinical Science, The Sahlgrenska Academy, University of Gothenburg, Gothenburg, Sweden
[2]Department of Health Care Sciences, Ersta Sköndal Bräcke University College, Stockholm, Sweden
[3]Centre for Sexology and Sexuality studies, Malmö Universitet, Malmo, Sweden

**Contributors** Guarantor of integrity of the entire study: GS. Study concepts and design: KB, GS, GD. Data analysis: LA, GD, KB, EE, GS. Statistical analysis: VS. Manuscript preparation: LA, KB, GD, EE. Manuscript editing: LA. All authors have read and approved the final manuscript.

**Funding** The authors have not declared a specific grant for this research from any funding agency in the public, commercial or not-for-profit sectors.

**Competing interests** None declared.

**Patient and public involvement** Patients and/or the public were not involved in the design, or conduct, or reporting, or dissemination plans of this research.

**Patient consent for publication** Not required.

**Ethics approval** All procedures in the study, which involved human participants, were in accordance with the ethical standards of the Regional Ethics Review Board in Gothenburg (D 686-10) and with the 1964 Declaration of Helsinki and its later amendments. The research was approved by the Regional Ethics Review Board in Gothenburg (D 686-10), Gothenburg University. Written informed consent was obtained from all individual participants included in the study.

**Provenance and peer review** Not commissioned; externally peer reviewed.

**Data availability statement** Data are available upon reasonable request.

**ORCID iD**
Linda Åkeflo http://orcid.org/0000-0002-6795-1092

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
