## [Reviewer comments · BMJ Open]

ARTICLE DETAILS

TITLE (PROVISIONAL)	Cohort profile: An observational longitudinal data collection of health aspects in a cohort of female cancer survivors with a history of pelvic radiotherapy, a population-based cohort in the Western Region of Sweden
AUTHORS	Åkeflo, Linda; Dunberger, Gail; Elmerstig, Eva; Skokic, Viktor; Steineck, Gunnar; Bergmark, Karin

VERSION 1 – REVIEW

REVIEWER	Courtney Stevens Dartmouth-Hitchcock Medical Center, Psychiatry
REVIEW RETURNED	13-Mar-2021

GENERAL COMMENTS	The title of the manuscript describes the objective of the study, but this language is not presented as clearly in the abstract or the body of the manuscript. Many of the references are over 10 years. if possible, the authors should add additional references that have been published more recently. The limitations section could be more clearly labeled and greater attention should be paid to critically examine the broad limitations of this work. Supplementary reporting: As this manuscript reports on the results of an observational study, I believe the STROBE checklist should be used and included with the submission. Standard of English. There are instances throughout the manuscript where phrasing is odd or redundant. Occasional issues with tense, word choice, and grammar. Overall the writing can be understood, but close editing would improve the readability of this manuscript. Other comments: What is the clinometric method? If it is important, please describe.
--

REVIEWER	Kimberly Keene University of Alabama at Birmingham, Radiation Oncology
REVIEW RETURNED	19-Mar-2021

GENERAL COMMENTS	The present study discusses a prospective population-based observational study set in Sweden, which aims to evaluate female patients with gynecological, anal, or rectal cancer who have received definitive-intent pelvic radiotherapy, especially with respect to late gastrointestinal, genitourinary, sexual, and lymphatic toxicity and to determine the effect of education and interventions relayed to
--

patients in a nurse-led clinic. The study approach and experimental design are appropriate. The study addresses important, original, and clinically relevant issues that could inform future studies or interventions. This paper discusses the clinical design and staffing needs for a survivorship clinic but it does not discuss any results of the patient interventions from this clinic. It describes the patient questionnaires used to define the extent of morbidity from prior cancer therapy but it also does not provide any results from the long term questionnaires from patients. This paper is mostly descriptive regarding the clinical setup, patient served and interventions offered. The manuscript states that the study was deemed ethically acceptable by an institutional review board.

Overall, the manuscript uses well-written English language prose. There are a few grammatical changes that could be considered. Specifically, on page 2, Abstract, Findings to date, line 18, suggest changing “report” to “reports.” In the introduction section on page 3, from lines 53-58, consider moving the following sentences to the methods section: “We collect data...participate in the study.” Again, on page 3, lines 58-60, consider revising the following sentence for clarity: “This paper outlines...then be reported.” Page 3, line 66, amend “finances” to “financed” to keep a consistent past tense. Page 3, line 68 remove comma splice. Page 4, line70, consider revising “working on issues concerning cancer survivorship” to be more specific. Page 4, line 71 add comma after “basic anatomy.” Page 4, line 71 add comma after “psychological.” Page 4, line 73 add comma after “intercurrent diseases.” Page 4, line79 substitute a colon for the semicolon. Page 4, line 81 remove comma. Page 4, line 83 revise “since completed radiotherapy” to “since the completion of radiotherapy.” Page 4, lines 83-85, consider revising “Exclusion criteria... Swedish at all” to “...cancer recurrence, inability to comprehend or answer the questionnaire, and poor proficiency in the Swedish language.” Page 4, line 88: revise this sentence to “...participate in the study, and if so asks them to give permission...” Please format any additional lists separated by commas in the manuscript consistently by using an Oxford comma. Page 7, line121 revise sentence to “...new strategies are suggested...” Page7, line 123-124 consider moving the following sentence to discussion “Cerna et al [17] ...tailor-made solutions.” Page 7, line 130, revise sentence to “blood tests such as electrolytes, blood counts, vitamin B12, and serum magnesium are...” Page 7, line 134 revise to “nurse’s.” Page 7, line 142 revise to “Therapy, which is sometimes...setting and at other times ...” Page 8, line 151 it is unclear what “regain body” means. Page 8, line 170 remove “for example.” Page 8, line 173-174 consider revising sentence; for example: “... is allowed for each question, in the event of two answers the procedure was to alternate entering the first provided answer and the second provided answer.” Page 9, line 191 add comma after “these.” Page 14, line 261 revise to “follow-up, which is consistent with results...” Page 14, line 261 replace the semicolon with a colon. Page 14, line 281 “previous had cancer” is unclear. Page 14, line 282 revise sentence to “...could be having symptoms which are too severe to travel...” Page 15, line 293 revise sentence to “It is worth mention that...” Page 15, line 296 replace “to” with “for.” Page 15, lines 298-301 consider revising conclusion; it is currently unclear. For example, revise sentence to “the interventions that are evaluated will contribute to the understanding of the manifestations of ...” and replace “future developments” with “further developments.”

	Figure 1 is helpful and clear. Table 1 is helpful and clear. Under sexual abuse revise “extend” to “extent” Figure 2 is helpful and clear. Under the section Education level revise “Collage” to “College.” Table 3 is helpful and clear. No formal statistical review is needed.
--	---

VERSION 1 – AUTHOR RESPONSE

Reviewer: 1 Dr. Courtney Stevens, Dartmouth-Hitchcock Medical Center Comments to the Author: The title of the manuscript describes the objective of the study, but this language is not presented as clearly in the abstract or the body of the manuscript.	Thank you for your comments. We have now clarified the objective of the study. The changes in the abstract can be found in line 5-6 and throughout the body of the manuscript, specifically in the introduction section, line 57-62.
Many of the references are over 10 years. If possible, the authors should add additional references that have been published more recently.	We agree and have updated the reference list. Three more references, published more recently, have now been added to the introduction, see Reference list no 7, 8 and 9. Furthermore, two more references have been added in “Findings to date”, see Reference list no 36, and 37.
The limitations section could be more clearly labeled and greater attention should be paid to critically examine the broad limitations of this work.	We agree with the reviewer that the limitations could be clearer. Yes, there are several limitations to this data collection and cohort that requires being further critically examined. Based on your comment, we have now added a clearer discussion of the limitations, outlined in lines 308-309, 312-313 and, 323-326.
Supplementary reporting: As this manuscript reports on the results of an observational study, I believe the STROBE checklist should be used and included with the submission.	We have now included the STROBE checklist with the submission.
Standard of English. There are instances throughout the manuscript where phrasing is odd or redundant. Occasional issues with tense, word choice, and grammar. Overall the writing can be understood, but close editing would improve the readability of this manuscript	Language revision of the manuscript has been carried out due to your appreciated suggestion and will hopefully improve the readability. Language revision has been carried out by a British language editor.
Other comments: What is the clinometric method? If it is important, please describe.	We have considered the clinometric methodology to be of great importance. A brief description of the method is now added, see line 317-321.
Reviewer: 2 Dr. Kimberly Keene, University of Alabama at Birmingham. Comments to the Author: The present study discusses a prospective population-based observational study set in Sweden, which aims to evaluate female patients	We appreciate the reviewer’s input and the positive comments about our work.

with gynecological, anal, or rectal cancer who have received definitive-intent pelvic radiotherapy, especially with respect to late gastrointestinal, genitourinary, sexual, and lymphatic toxicity and to determine the effect of education and interventions relayed to patients in a nurse-led clinic. The study approach and experimental design are appropriate. The study addresses important, original, and clinically relevant issues that could inform future studies or interventions. This paper discusses the clinical design and staffing needs for a survivorship clinic but it does not discuss any results of the patient interventions from this clinic. It describes the patient questionnaires used to define the extent of morbidity from prior cancer therapy but it also does not provide any results from the long term questionnaires from patients. This paper is mostly descriptive regarding the clinical setup, patient served and interventions offered. The manuscript states that the study was deemed ethically acceptable by an institutional review board.	
Overall, the manuscript uses well-written English language prose. There are a few grammatical changes that could be considered. Specifically, on page 2, Abstract, Findings to date, line 18, suggest changing “report” to “reports.” In the introduction section on page 3, from lines 53-58, consider moving the following sentences to the methods section: “We collect data...participate in the study.” Again, on page 3, lines 58-60, consider revising the following sentence for clarity: “This paper outlines...then be reported.” Page 3, line 66, amend “finances” to “financed” to keep a consistent past tense. Page 3, line 68 remove comma splice. Page 4, line70, consider revising “working on issues concerning cancer survivorship” to be more specific. Page 4, line 71 add comma after “basic anatomy.” Page 4, line 71 add comma after “psychological.” Page 4, line 73 add comma after “intercurrent diseases.” Page 4, line79 substitute a colon for the semicolon. Page 4, line 81 remove comma. Page 4, line 83 revise “since completed radiotherapy” to “since the completion of radiotherapy.” Page 4, lines 83-85, consider revising “Exclusion criteria... Swedish at all” to “...cancer recurrence, inability to comprehend or answer the questionnaire, and poor proficiency	Thank you for pointing out the grammatical weaknesses. The changes made throughout the manuscript according to your appreciated suggestions have hopefully improved the manuscript.

in the Swedish language.” Page 4, line 88: revise this sentence to “...participate in the study, and if so asks them to give permission...”	
...” Please format any additional lists separated by commas in the manuscript consistently by using an Oxford comma.”	Thank you for your remark, an Oxford comma is now used consistently throughout the manuscript according to your suggestion.
Page 7, line121 revise sentence to “...new strategies are suggested...” Page7, line 123-124 consider moving the following sentence to discussion “Cerna et al [17] ...tailor-made solutions.” Page 7, line 130, revise sentence to “blood tests such as electrolytes, blood counts, vitamin B12, and serum magnesium are...” Page 7, line 134 revise to “nurse’s.” Page 7, line 142 revise to “ Therapy, which is sometimes...setting and at other times ...” Page 8, line 151 it is unclear what “regain body” means. Page 8, line 170 remove “for example.” Page 8, line 173-174 consider revising sentence; for example: “... is allowed for each question, in the event of two answers the procedure was to alternate entering the first provided answer and the second provided answer.” Page 9, line 191 add comma after “these.” Page 14, line 261 revise to “follow-up, which is consistent with results...” Page 14, line 261 replace the semicolon with a colon. Page 14, line 281 “previous had cancer” is unclear. Page 14, line 282 revise sentence to “...could be having symptoms which are too severe to travel...” Page 15, line 293 revise sentence to “It is worth mention that...” Page 15, line 296 replace “to” with “for.” Page 15, lines 298-301 consider revising conclusion; it is currently unclear. For example, revise sentence to “the interventions that are evaluated will contribute to the understanding of the manifestations of ...” and replace “future developments” with “further developments.”	We thank you for your detailed comments. We have made the changes according to your suggestions and hopefully the manuscript has improved.
Figure 1 is helpful and clear. Table 1 is helpful and clear. Under sexual abuse revise “extend” to “extent”. Figure 2 is helpful and clear. Under the section Education level revise “Collage” to “College.” Table 3 is helpful and clear. No formal statistical review is needed.	We thank you for your comments. We have made these changes.

VERSION 2 – REVIEW

REVIEWER	Courtney Stevens Dartmouth-Hitchcock Medical Center, Psychiatry
REVIEW RETURNED	01-Jul-2021
GENERAL COMMENTS	This manuscript is much improved and I have no further comments.